# Human macular Müller cells rely more on serine biosynthesis to combat oxidative stress than those from the periphery

Ting Zhang[1†], Ling Zhu[1†*], Michele C Madigan[1,2], Wei Liu[3], Weiyong Shen[1], Svetlana Cherepanoff[4], Fanfan Zhou[5], Shaoxue Zeng[1], Jianhai Du[6,7], Mark C Gillies[1]

[1]Save Sight Institute, Sydney Medical School, Faculty of Medicine and Health, The University of Sydney, Sydney, Australia; [2]School of Optometry and Vision Sciences, University of New South Wales, Sydney, Australia; [3]Clinical Genomics Laboratory, Sidra Medicine, Doha, Qatar; [4]Department of Anatomical Pathology, St Vincent's Hospital, Darlinghurst, Australia; [5]Faculty of Pharmacy, The University of Sydney, Sydney, Australia; [6]Department of Ophthalmology, West Virginia University, Morgantown, United States; [7]Department of Biochemistry, West Virginia University, Morgantown, United States

**Abstract** The human macula is more susceptible than the peripheral retina to developing blinding conditions such as age-related macular degeneration, diabetic retinopathy. A key difference between them may be the nature of their Müller cells. We found primary cultured Müller cells from macula and peripheral retina display significant morphological and transcriptomic differences. Macular Müller cells expressed more phosphoglycerate dehydrogenase (PHGDH, a rate-limiting enzyme in serine synthesis) than peripheral Müller cells. The serine synthesis, glycolytic and mitochondrial function were more activated in macular than peripheral Müller cells. Serine biosynthesis is critical in defending against oxidative stress. Intracellular reactive oxygen species and glutathione levels were increased in primary cultured macular Müller cells which were more susceptible to oxidative stress after inhibition of PHGDH. Our findings indicate serine biosynthesis is a critical part of the macular defence against oxidative stress and suggest dysregulation of this pathway as a potential cause of macular pathology.
DOI: https://doi.org/10.7554/eLife.43598.001

**\*For correspondence:**
ling.zhu@sydney.edu.au

[†]These authors contributed equally to this work

**Competing interests:** The authors declare that no competing interests exist.

## Introduction

The macula, a specialized region located at the posterior pole of the primate retina, has the greatest density of cone photoreceptors for the highest visual acuity (*Sharon et al., 2002*). The impact of common pathologies such as age-related macular degeneration (AMD), diabetic retinopathy (DR) and macular telangiectasia type 2 (MacTel) are most devastating at the macula, leading to vision loss and blindness. Understanding the unique biochemical and anatomic specializations of the macula may provide new ways to prevent and treat these diseases.

A major difference between the macula and peripheral retina is the density of the retinal neurons. Although the macula occupies less than 1.4% of the retina area, it contains ~8.4% of cones, ~3.4% of rods and ~60% of the ganglion cells (*Curcio and Allen, 1990*). The ratio of cones to rods is much higher in the macula (1:8) than that in the peripheral retina (1:20) (*Curcio et al., 1990*). The high density of neurons in the macula is associated with a high metabolic rate and increased levels of oxidative stress (*Handa, 2012*).

Müller cells, the major glial cells of retina, are responsible for retinal redox homeostasis and metabolic support of retinal neurons (*Reichenbach and Bringmann, 2013*). They control the composition of retinal extracellular fluids by mediating the transcellular transport of ions, water and bicarbonate (*Bringmann et al., 2005*; *Kofuji and Newman, 2004*; *Newman, 1996*). They may play an important role in retinal glucose metabolism and provide substrates to photoreceptors, one of the most metabolically active cell types in the body (*Poitry-Yamate et al., 1995*). Müller cells also promote photoreceptor survival by providing neurotrophic substances and removing metabolic end products (*Poitry et al., 2000*). Whether these functional requirements of Müller cells differ in the macula and the retinal periphery is unknown.

Müller cells in the macula display unique morphological features. The apical processes of all Müller cells envelop the photoreceptor cell bodies within the outer nuclear layer as the outer limiting membrane, extending distally to form the 'fibre baskets' (*Bringmann et al., 2006*). Müller cells in the periphery share most of these features. Foveal Müller cells are Z-shaped: their processes run vertically from the inner limiting membrane (ILM) to the inner nuclear layer (INL), then horizontally away from the fovea within Henle's fibre layer (HFL) then vertically again to the outer limiting membrane (OLM). A specialized, cone-shaped zone of Müller cells described in the central and inner part of the fovea has recently received renewed attention (*Gass, 1999*; *Yamada, 1969*).

Macular and peripheral Müller cells may also differentially express several functional proteins. Müller cells from the macula and the periphery both express CD117 but only peripheral Müller cells express CD44 (*Too et al., 2017*). Macular Müller cells also express more Aquaporin-4 than peripheral Müller cells, which may contribute to a specialized 'glymphatic system' of the macula (*Daruich et al., 2018*).

The importance of Müller cells for retinal homeostasis suggests that their dysfunction contributes to many retinal diseases. Retinal oedema and neuronal degeneration, the common features of many retinal diseases, may occur secondary to abnormal retinal glutamate metabolism and ionic disturbances caused by Müller cell dysfunction (*Bringmann et al., 2006*). Müller cell dysfunction has been implicated in MacTel) (*Powner et al., 2010*) and in a form of autoimmune retinopathy (*Peek et al., 1998*). Any primary or secondary impairment of Müller cells may cause or at least aggravate the degeneration of retinal neurons by increasing their susceptibility to noxious stimuli. Recent report suggests that Müller cells are the major sites in the retina of de novo synthesis of serine and glutathione which support retinal neurons and protect them from oxidative stress (*Zhang et al., 2018*). Therefore, we investigated molecular and functional differences of Müller cells in the macula and peripheral regions of retina, with particular interest in serine and glycine metabolism.

## Results

### Isolation, culturing and validation of primary Müller cells from human macula and peripheral retina

We isolated macular and mid-peripheral retinal pieces from healthy human donors' eyes and cultured the Müller cells as previously described (*Zhu et al., 2015*) (*Figure 1A*). The anatomical boundary of the macula is commonly recognized as a region approximately 5.5-mm-diameter, centred at the fovea (*Chopdar et al., 2003*). We thus trephined 5 mm diameter disks centred at the fovea and similarly sized pieces from mid-peripheral retina (4.5 mm to 7.5 mm from the fovea) (*Mori et al., 2016*). After 4–6 weeks in culture, Müller cells that grew out from the macular (macular-human primary Müller cells [M-huPMCs]) and peripheral (peripheral-human primary Müller cells [P-huPMCs]) retinal pieces had distinct cell morphologies (*Figure 1D & F*). M-huPMCs were small, spindle- or stellate-shaped cells with a low cytoplasm/nucleus ratio (*Figure 1B & C*); the P-huPMCs had large cell bodies, multiple cytoplasmic processes and a higher cytoplasm/nucleus ratio (*Figure 1B & E*). Both M-huPMCs and P-huPMCs without sub-culturing (P0) demonstrated positive immunolabeling for four Müller-cell-specific markers: Glial fibrillary acidic protein (GFAP) (*Figure 1G–H*), carbonic anhydrase II (*Figure 1I–J*), SOX9 (*Figure 1K–L*) (*Song et al., 2013*) and Cellular Retinaldehyde Binding Protein (CRALBP) (*Figure 1—figure supplement 1*). GFAP is a non-specific response of Müller cell stress which they usually express in cell culture (*Augustine et al., 2018*). This confirmed that these two morphologically distinct cell populations are Müller cells.

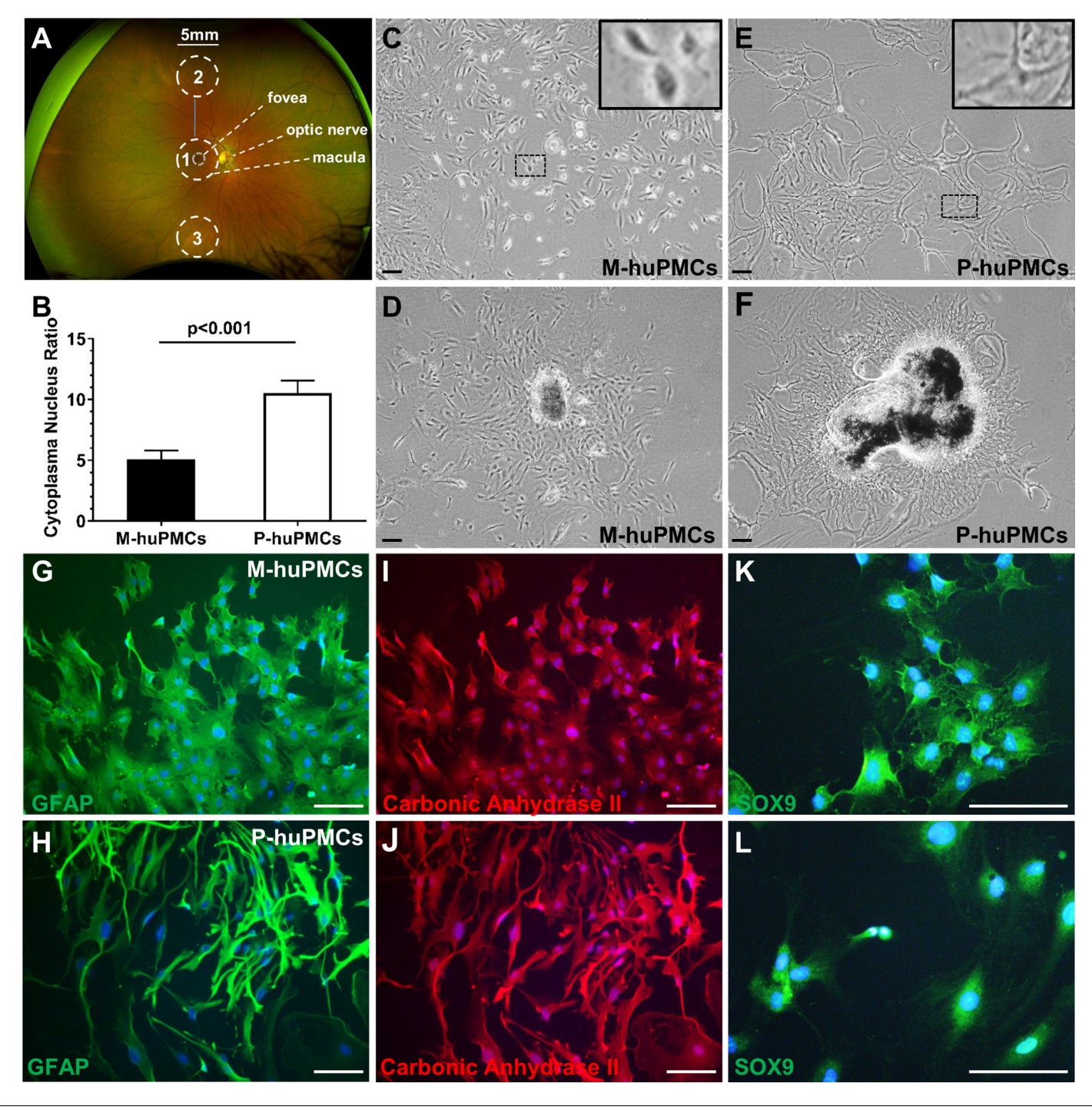

**Figure 1.** Morphology of primary macular and peripheral Müller cells. (**A**) A wide-field fundus image showing the retinal areas used for primary Müller cell culture. Retinal tissue at Area one was used to derive primary Müller cells from the macula while the punches from Areas 2 and 3 were pooled to derive primary Müller cells from the peripheral retina. (**B**) Cytoplasm/nucleus ratio of primary Müller cells from the macula (M-huPMCs) and peripheral (P-huPMCs) human retina (n = 8). (**C-F**) Bright field images of huPMCs isolated from the macula (**C-D**) or peripheral (**E-F**) retina. (**D**) Primary Müller cells that grew out from macular retinal piece. (**F**) Primary Müller cells that grew out from peripheral retinal piece. (**G-L**) M-huPMCs (upper panel) and P-huPMCs (lower panel) (P0) without sub-culturing expressed three specific protein markers of Müller cells: GFAP (**G-H**), carbonic anhydrase II (**I-J**) and SOX9 (**K-L**). Black scale bar = 200 µM, white scale bar = 100 µM.

DOI: https://doi.org/10.7554/eLife.43598.002

The following source data and figure supplement are available for figure 1:

*Figure 1 continued on next page*

*Figure 1 continued*

**Source data 1.** Source data for *Figure 1B*.
DOI: https://doi.org/10.7554/eLife.43598.004
**Figure supplement 1.** CRALBP immunostaining of M-huPMCs and P-huPMCs.
DOI: https://doi.org/10.7554/eLife.43598.003

## Transcription profiles of macular and peripheral Müller cells

RNA-sequencing analysis of matched M-huPMCs (n = 8) vs P-huPMCs (n = 8) from each individual retina of eight donors revealed significant differences between the two cell types. Reads expression values of all 16 samples suggested a highly similar gene expression pattern (*Figure 2A*). Scatter plots show the correlation of the gene expression profiles between M-huPMCs and P-huPMCs (*Figure 2B*). In *Figure 2C* heatmap of gene expressions of M-huPMCs and P-huPMCs (R, DESeq2) is used to visualize the top 100 genes with the smallest q-values. Pairwise sample distances were estimated using the Euclidean distance and sample clustering was done with the Ward algorithm. Principal component analysis (PCA) shows the clustering of M-huPMCs and P-huPMCs (*Figure 2D*). A total of 7588 differentially expressed genes (DEGs) were identified in Müller cells isolated from human macula and peripheral retina with a 1.5-fold or more increase or decrease and with an FDR corrected p-value<0.05 (*Figure 2F*, Supplementary S1 and S2). Bland–Altman (MA) plot and Volcano plot show the differentially expressed genes in M-huPMCs and P-huPMCs (*Figure 2E–F*). Based on gene ontology (GO) enrichment analysis (ClusterProfiler), these upregulated genes were significantly associated with numerous GO Biological process terms, including 'response to oxygen levels', 'response to decreased oxygen levels' and 'response to hypoxia' (*Figure 2G*).

RNA sequencing found that both the M-huPMCs and P-huPMCs strongly expressed (high RPKM number [Reads Per Kilobase of transcript per Million mapped reads]) gene markers for the Müller cell: vimentin, glutamine synthetase (GS), CRALBP, clusterin, carbonic anhydrase II and GFAP (*Table 1*) and negligibly expression (low RPKM number) of gene markers of the other retinal cell types: rhodopsin for photoreceptors; mGluR6 for the bipolar cells, syntaxin 1A for the amacrine cells, Iba1 for the microglia cells and PECAM1 for vascular endothelial cells (*Table 1*). These findings are consistent with Müller cells retaining their original genotype in primary tissue culture. Notably, the expression of Müller cell markers was higher in the P-huPMCs compared to M-huPMCs.

## Differential expression of genes related to the de novo serine synthesis pathway in Müller cells isolated from the human macula and peripheral retina

MacTel is a retinal disease which may be caused primarily by dysfunction of macular Müller cells. A recent study reported that the de novo serine metabolic pathway may play an important role in MacTel through defects of phosphoglycerate dehydrogenase (PHGDH), a key rate-limiting enzyme in de novo serine metabolism (*Scerri et al., 2017*). We have reported that PHGDH is predominantly expressed in human Müller cells and that inhibition of PHGDH aggravates mitochondrial and oxidative stress (*Zhang et al., 2018*). We found in the present study that the expression of key genes related to de novo serine synthesis (e.g. PHGDH, PSAT1 and SHMT2) was higher in M-huPMCs than in P-huPMCs (*Table 2*).

Western blot studies found that M-huPMCs expressed significantly more PHGDH compared to P-huPMCs (*Figure 3A–B*, normalized to β-actin, n = 4 donor retinas). We further explored the topographical protein expression of PHGDH using a series of 2 mm fresh retinal tissue punches at different locations across the human retina (*Figure 3C*). We found that macula expressed significantly more PHGDH (*Figure 3D–F*, normalized to α/β-Tubulin or *Figure 3—figure supplement 1*, normalized to total loaded retinal protein) than punches from the peripheral retina, although there were inter-individual variations between the five retinas. We believe that PHGDH expression from the retina largely reflects its expression by Müller cells since they are the main source of PHGDH in the human retina (*Zhang et al., 2018*).

Immunofluorescent staining of human macular (*Figure 4A–H*) and peripheral (*Figure 4I–P*) retina from the same eye revealed co-localization of CRALBP (red, a Müller cell marker) and PHGDH (green). This is consistent with our previous study that PHGDH in the retina is predominantly

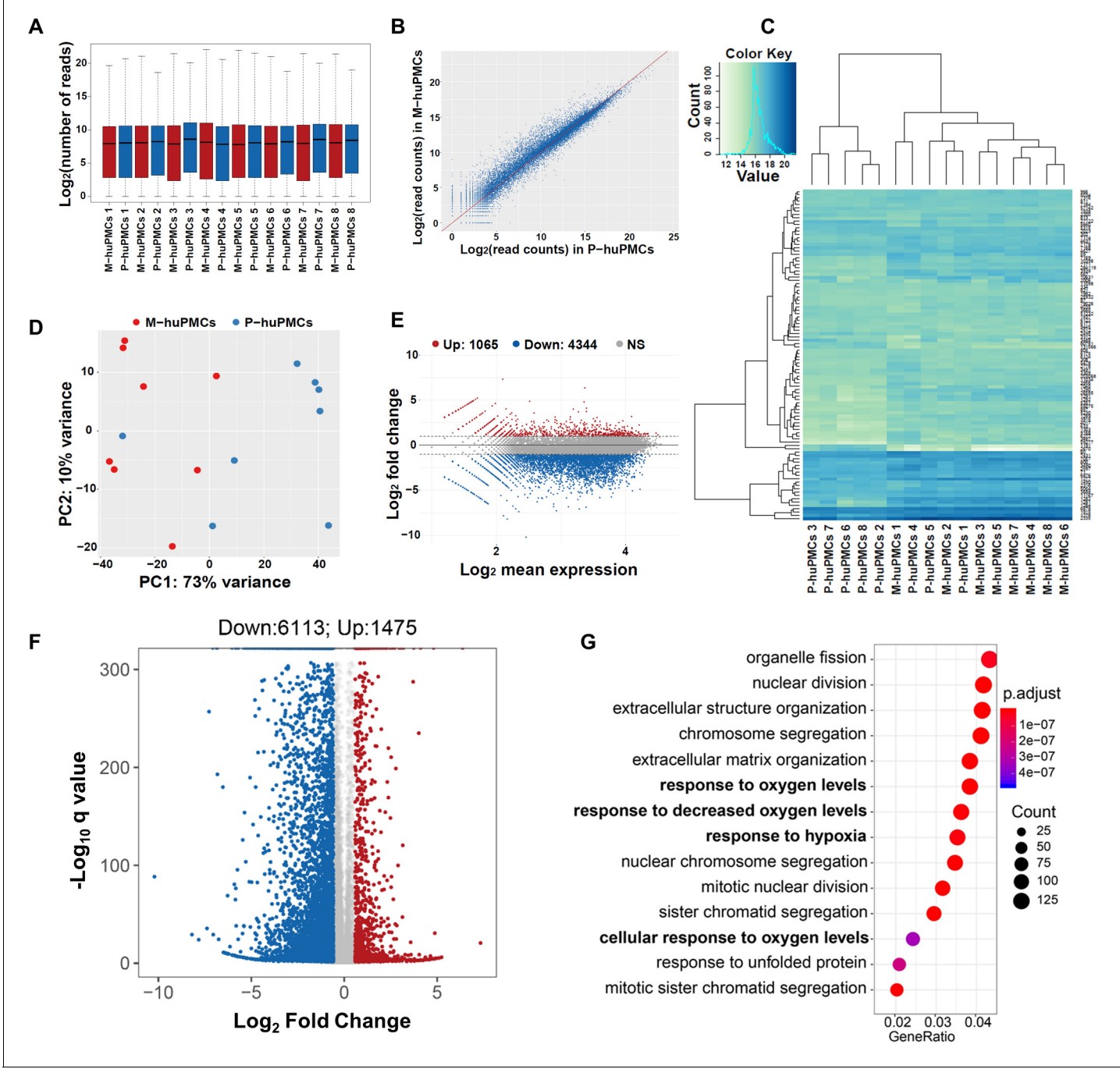

**Figure 2.** Transcription profiles of macular and peripheral Müller cells. (**A**) Boxplot of the Reads expression values of the M-huPMCs and P-huPMCs cultured from eight donor retinas. (**B**) Scatter Plot for correlations of genes expressed in the M-huPMCs and P-huPMCs. (**C**) Heatmap of differentially expressed genes between the M-huPMCs and P-huPMCs of eight donors shows the top 100 genes with the smallest q-values. (**D**) Principal component analysis (PCA) was performed with the RNA-seq data derived from the M-huPMCs and P-huPMCs. (**E**) Bland–Altman (MA) plot of differentially expressed genes in the M-huPMCs and P-huPMCs. (**F**) Volcano plot of differentially expressed genes in the M-huPMCs and P-huPMCs. The log 10 (q values) were plotted against the log 2 (Fold Change) in gene expression. The genes upregulated (n = 1475) 1.5-fold or more and with an FDR corrected p-value<0.05 were depicted as red dots; while the genes downregulated (n = 6113) by 1.5-folds or more and with a FDR corrected p-value<0.05 were depicted as blue dots. (**G**) Gene ontology (GO) analysis of upregulated genes in M-huPMCs relative to P-huPMCs. The top 14 GO terms in the biological process category are displayed and ordered by enriched gene number and adjusted p value.

DOI: https://doi.org/10.7554/eLife.43598.005

**Table 1.** Gene expression of retinal cells marker in identified cells from macula and peripheral retina

| Marker | Gene name | Gene ID | Length | M-rpkm | P-rpkm | Log$_2$ratio(P/M) | Regulation(P/M) | p-value |
|---|---|---|---|---|---|---|---|---|
| Müller cell | Vimentin | 7431 | 2151 | 2923.82 | 3492.22 | 0.26 | up | <2.22E-308 |
| | Glutamine synthetase | 2752 | 4737 | 23.46 | 95.31 | 2.02 | up | <2.22E-308 |
| | RLBP1 (CRALBP) | 6017 | 1752 | 4.50 | 34.79 | 2.95 | up | <2.22E-308 |
| | CLU clusterin | 1191 | 3012 | 78.80 | 763.72 | 3.28 | up | <2.22E-308 |
| | Carbonic anhydrase II | 760 | 1666 | 15.52 | 138.45 | 3.16 | up | <2.22E-308 |
| | GFAP | 2670 | 3097 | 31.87 | 944.58 | 4.89 | up | <2.22E-308 |
| Other cell types | | | | | | | | |
| Photoreceptors | RHO rhodopsin | 6010 | 2768 | 0.01 | 0.05 | 2.13 | up | 1.43E-07 |
| Biopolar cells | mGluR6 | 2916 | 6143 | 0.03 | 0.31 | 3.2 | up | 3.13E-134 |
| Amacrine cells | STX1A syntaxin 1A | 6804 | 2138 | 4.35 | 3.89 | −0.16 | down | 3.43E-06 |
| Microglia | Iba1 | 199 | 1491 | 1.44 | 3.67 | 1.35 | up | 3.99E-129 |
| Endothelial | PECAM1 | 5175 | 6831 | 0.25 | 0.72 | 1.52 | up | 1.21E-138 |

DOI: https://doi.org/10.7554/eLife.43598.006

expressed in the Müller cells(*Zhang et al., 2018*). In contrast to the peripheral retina (*Figure 4M–P*), the macula has multiple layers of Müller cell nuclei in the inner nuclear layer and strongly expresses PHGDH in Henle fibre layer (*Figure 4E–H*).

## Metabolic differences between the macular and peripheral Müller cells

To evaluate the metabolism of differential expression of PHGDH in the macular and peripheral Müller cells, we treated M-huPMCs and P-huPMCs from four different donors with $^{13}$C-glucose to measure the metabolic flux of serine and glycine from glucose (*Figure 5A–B*). Following the treatment, different $^{13}$C labelled serine isotopologue (M0, M1, M2, and M3) and glycine isotopologue (M0, M1 and M2) were quantified with gas chromatography–mass spectrometry (GC-MS). After 3 hr labeling, around 18% of cellular serine (average of both M-huPMCs and P-huPMCs) was replaced by $^{13}$C-glucose (*Figure 5C*). M-huPMCs had higher $^{13}$C labelled serine (24%) and $^{13}$C-labeled glycine (12%) than of P-huPMCs (13% and 5% respectively, *Figure 5C*). We did not observe a significant change in the relative level of total serine or glycine in M-huPMCs and P-huPMCs (*Figure 5—figure supplement 1*).

We also treated M-huPMCs and P-huPMCs with $^{13}$C-labeled serine to trace how human primary Müller cells use exogenous serine. After treatment with $^{13}$C-serine for 24 hr, we found that most $^{13}$C-serine was converted to $^{13}$C-glycine but we found no other glucose intermediates, in contrast to previous reports in cancer cells (*Amelio et al., 2014*). A modest but statistically significantly higher level of $^{13}$C-labeled M2 glycine (70% vs 59%) suggests that more serine is converted to glycine in M-huPMCs than in P-huPMCs. However, less $^{13}$C-labeled M1 glycine, M1 serine and M2 serine in M-huPMCs indicates that there is a lower rate of reverse serine/glycine reactions through SHMT occurring in these cells than in P-huPMCs (*Figure 5D*).

The expression of PHGDH is not only important to the de novo serine synthesis metabolic pathway, but also closely related to cellular glycolytic and mitochondrial functions. We further assessed

**Table 2.** Serine synthesis pathway-related gene expression in Müller cell isolated from macula and peripheral retina.

| Gene name | Gene ID | Length | M-rpkm | P-rpkm | Log$_2$ratio (M/P) | Regulation (M/P) | p-value |
|---|---|---|---|---|---|---|---|
| PHGDH | 26227 | 2021 | 60.36 | 40.09 | 0.59 | up | <2.22E-308 |
| PSAT1 | 29968 | 2221 | 101.15 | 79.10 | 0.35 | up | <2.22E-308 |
| PSPH | 5723 | 2142 | 12.51 | 14.20 | −0.18 | down | 2.56E-21 |
| SHMT1 | 6470 | 2540 | 3.72 | 7.05 | −0.92 | down | 2.99E-228 |
| SHMT2 | 6472 | 2357 | 64.83 | 61.14 | 0.08 | up | 1.27E-23 |

DOI: https://doi.org/10.7554/eLife.43598.007

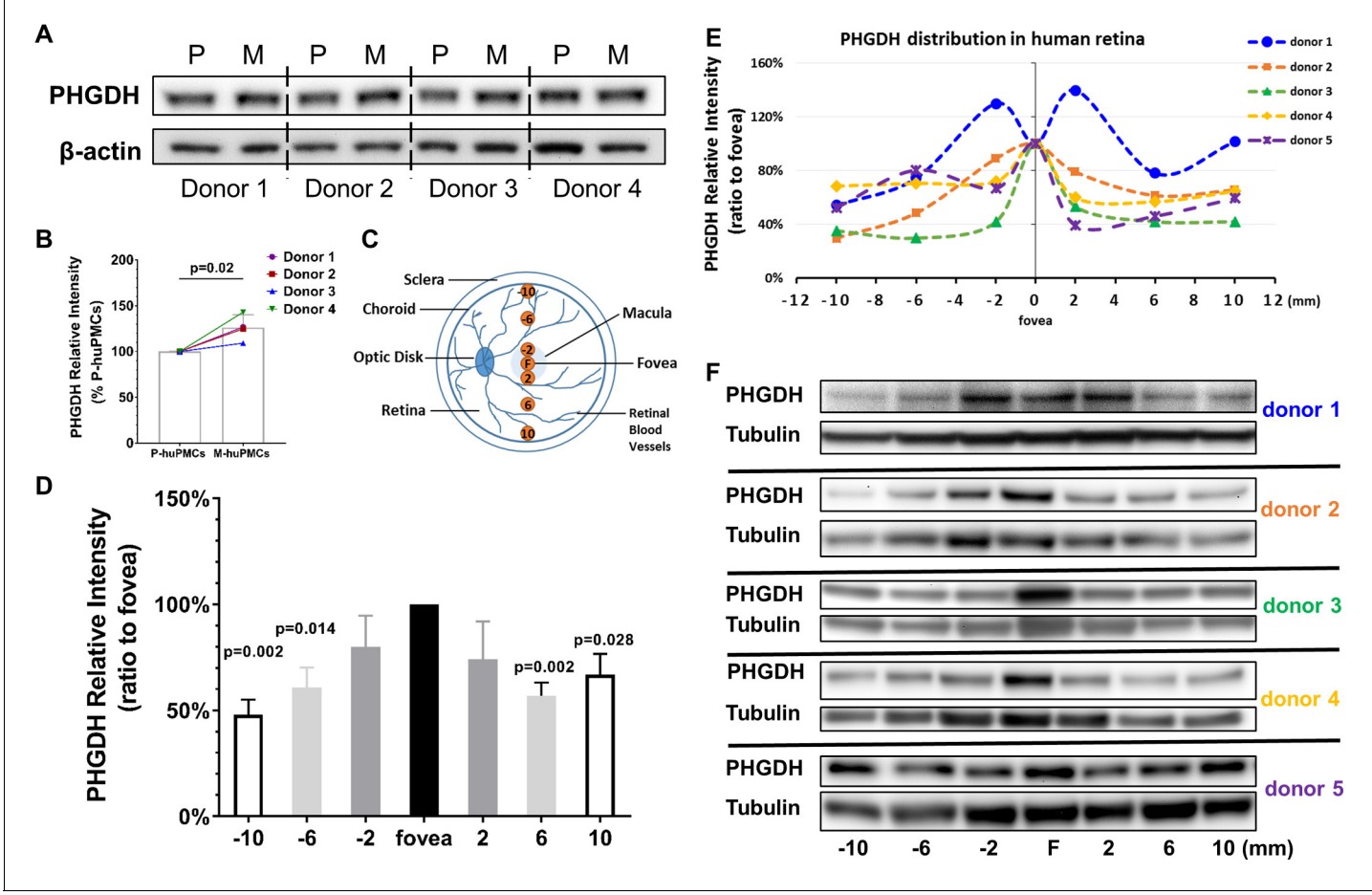

**Figure 3.** PHGDH expression in cultured Müller cells and retinal tissue from macula and periphery. (A) Representative immunoblot showing PHGDH protein expression in primary cultures of human Müller cells from macula and peripheral retina; (B) Quantitative analysis of PHGDH relative to β-actin for immunoblots (n = 4 donor retinas); (C) Schema of trephined retina area used for Western blotting shown in (F); (D-E) Quantitative analysis of the expression of PHGDH correlated to different areas of human retina (five donors); (F) Expression of PHGDH in different retinal locations shown using western blotting (n = 5 donors).

DOI: https://doi.org/10.7554/eLife.43598.008

The following source data and figure supplement are available for figure 3:

**Source data 1.** Source data for *Figure 3B,D,E,F*.
DOI: https://doi.org/10.7554/eLife.43598.010

**Figure supplement 1.** PHGDH expression in the macula and peripheral retina.
DOI: https://doi.org/10.7554/eLife.43598.009

the glycolytic and mitochondrial functions of M-huPMCs and P-huPMCs using Seahorse Assay. We found that M-huPMCs had significantly higher glycolytic capacity and reserve than P-huPMCs (*Figure 5E–F*). M-huPMCs also had significantly higher basal mitochondrial respiration and ATP production but lower spare respiratory capacity than P-huPMCs (*Figure 5G–H*).

## Susceptibility of the M-huPMCs and P-huPMCs to oxidative stress after inhibition of PHGDH

To further explore the role of PHGDH in Müller cells from different regions, we used a PHGDH specific inhibitor, CBR-5884, to assess the different responses of Müller cells from different retinal regions. We first validated the inhibition efficiency of CBR-5884 in human primary Müller cells and found that treatment of CBR-5884 in huPMCs for 2 hr resulted in 61% and 42% reductions in serine and glycine levels, respectively (*Figure 6A*). We also found that M-huPMCs and P-huPMCs had similar levels of cell viability and cytotoxicity after exposure to CBR-5884 (30 μM) for 6 hr. M-huPMCs

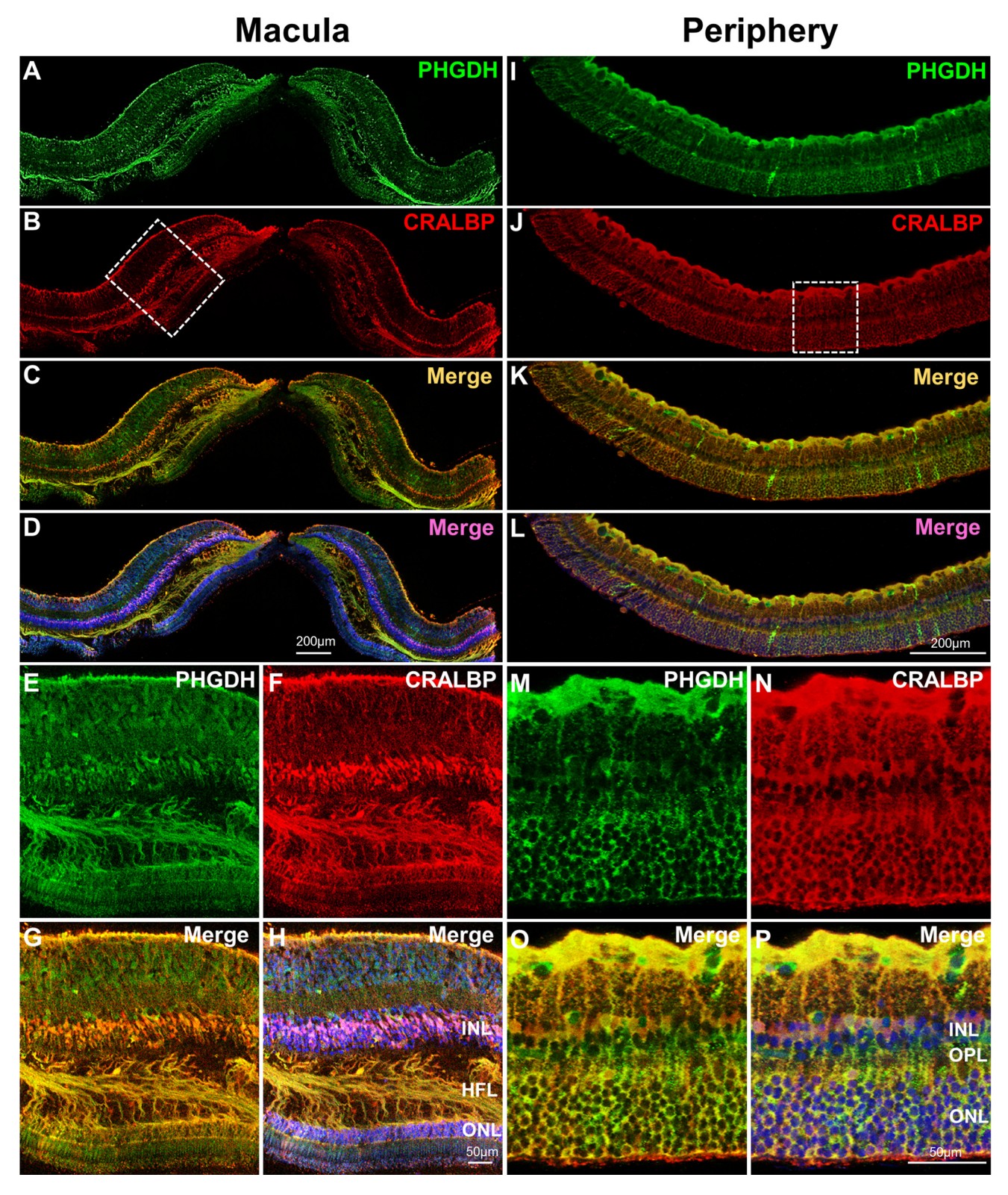

**Figure 4.** Immunofluorescent staining of PHGDH in human retina. Representative images of immunofluorescence visualization of PHGDH (green) and CRALBP (red, a Müller cell marker) in human macular (**A-H**) and peripheral retina (**I-P**). **E-H** Field enlarged image from **B** (white-dotted box) in macula; **M-P**. Enlarged images from **J** (white-dotted box) in peripheral retina. The specific immunoreactivity of PHGDH antibody was verified in PHGDH knockdown in MIO-M1 cells (*Figure 4—figure supplement 1*).

*Figure 4 continued on next page*

*Figure 4 continued*

DOI: https://doi.org/10.7554/eLife.43598.011

The following figure supplement is available for figure 4:

**Figure supplement 1.** Verification of PHGDH specific immunoreactivity by PHGDH knockdown in MIO-M1 cells.

DOI: https://doi.org/10.7554/eLife.43598.012

had significantly lower cell viability and higher cytotoxicity levels than P-huPMCs when they were exposed to exogenous oxidative stress (100 µM H$_2$O$_2$) (*Figure 6B–C*). Inhibiting PHGDH using CBR-5884 when cells were exposed to oxidative stress led to more pronounced Müller cells cytotoxicity in the macula compared to the periphery.

Since de novo serine synthesis is closely related to the production of glutathione (GSH, the major antioxidant in retina), we speculated that the greater susceptibility of Müller cells to oxidative stress when PHGDH was inhibited, was due to disturbance of the balance between GSH and ROS. We measured GSH and ROS levels in M-huPMCs and P-huPMCs under oxidative stress with or without PHGDH inhibition using flow cytometry (*Figure 6D–I*). We found significantly higher intracellular levels of GSH in M-huPMCs than in P-huPMCs (*Figure 6F*). When the cells were also subjected to mild oxidative stress (100 µM H$_2$O$_2$), P-huPMCs had a more pronounced decrease of GSH levels and elevated ROS levels than M-huPMCs. Inhibition of PHGDH using CBR-5884 induced a more significant GSH reduction in M-huPMCs than in P-huPMCs; ROS levels significantly increased in both M-huPMCs and P-huPMCs (*Figure 6F–I*, normalized to control of P-huPMCs or *Figure 6—figure supplement 1*, normalized to P-huPMCs/M-huPMCs own control).

## Discussion

To understand macula pathology, we studied the differences between Müller cells from the peripheral and central retina in primary culture. We found that Müller cells from the macula showed a morphologically distinct response to primary tissue culture compared to those from the peripheral retina. Previous reports have highlighted the morphological differences between Müller cells in the human macula and the retinal periphery in vivo (*Bringmann et al., 2018*). In situ Müller cells in the foveal region display a Z shape; their elongated processes run with the cone axons in Henle's fibre layer, most conspicuously in the parafovea (*Lujan et al., 2011*). In culture, we found that Müller cells isolated from macula were small spindle to stellate shaped cells with lower cytoplasm/nucleus ratios and shorter processes, while the Müller cells from peripheral retina had larger cell bodies, multiple cytoplasmic processes, higher cytoplasm/nucleus ratios. This might be due to the different responses of Müller cells to in vitro culture conditions. These phenotypic differences were retained in the two sub-populations of Müller cells up to the second passage (P2), after which cells showed evidence of division more towards immortalized 'MIO-M1'-shaped cells. This suggested that these two populations of cells can maintain distinct characteristics up to the second passage, after which they revert to a common phenotype. Therefore, all our following experiments were strictly controlled to use the primary cells up to two passages. Loss of surrounding cells and other in vivo conditions is an acknowledged limitation of tissue culture; however, the different responses of macular *vs.* peripheral Müller cells to the same in vitro conditions suggests a fundamental difference in mechanisms of survival and homeostasis.

While the two types of cells had distinct morphological appearances, immunofluorescent studies confirmed that Müller cells from the macula and retinal periphery both expressed the same range of Müller cell markers (*Figure 1*); thus, differences in the cytoskeletal and other proteins underpinning these distinct morphologies are yet to be characterized. Our RNA-seq analysis significantly revealed differential gene expression patterns in Müller cells from the macula and peripheral retina. However, both populations of cells strongly expressed Müller cell markers and only weakly expressed markers of other retinal cell types. Interestingly, peripheral Müller cells expressed Müller cell markers more strongly than those from the macula, perhaps in keeping with their ability to maintain a more 'Müller-like' morphology in primary culture. Müller cells from the peripheral retina of humans may be more like Müller cells in other species, most of which do not have a macula. It has been reported

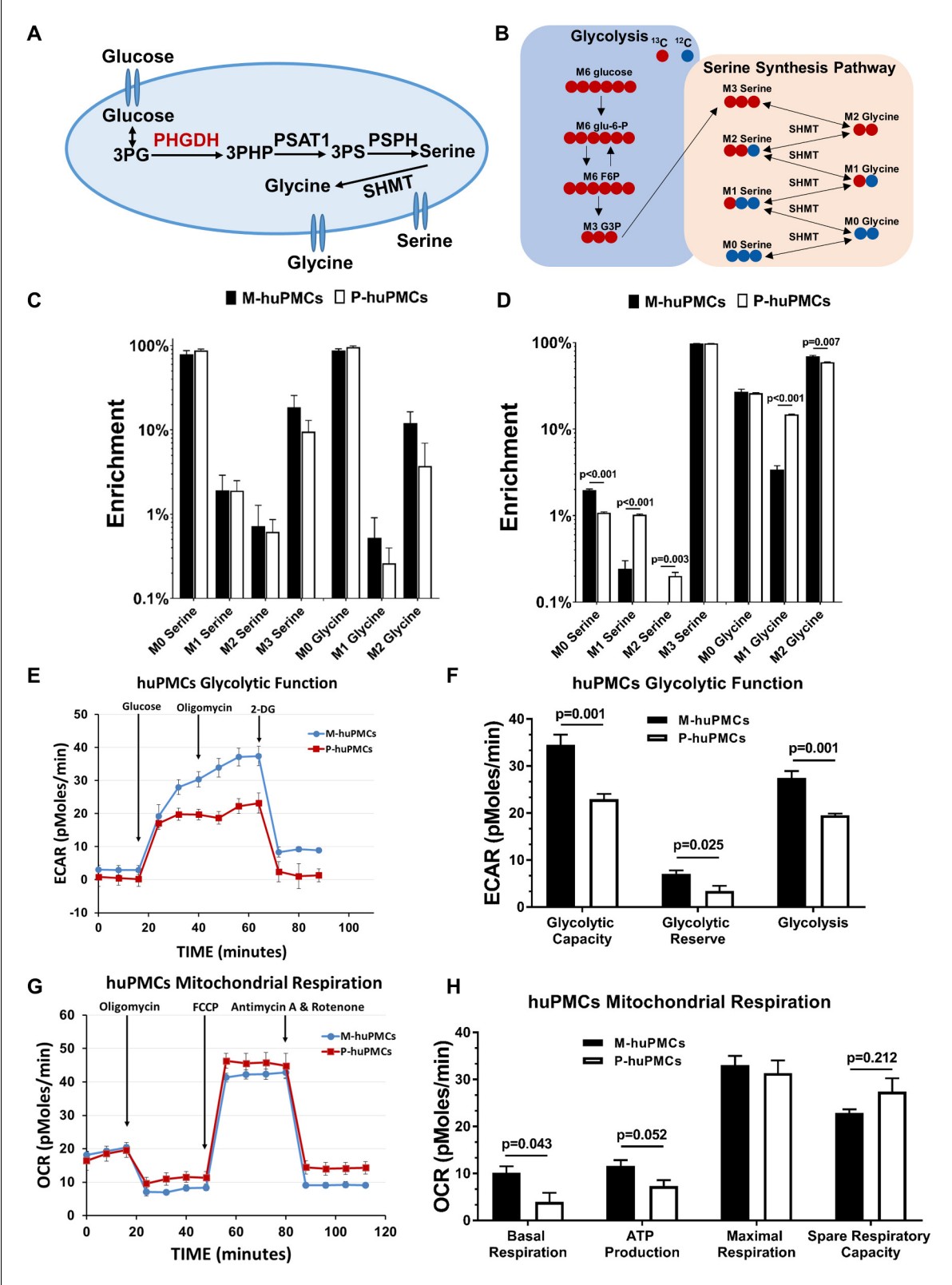

**Figure 5.** Metabolic differences between the macular and peripheral Müller cells. (A) de novo serine/glycine synthesis pathway. 3PG: 3-phosphoglycerate, PHGDH: 3-phosphoglycerate dehydrogenase, PSAT1: phosphoserine aminotransferase 1, PSPH: phosphoserine phosphatase, SHMT: serine hydroxymethyltransferase. (B) Simplified schematic of steps in de novo serine/glycine synthesis, showing [13]C labeling patterns resulting from [13]C6 glucose substrate. Red fills indicate [13]C atoms. (C–D) [13]C-serine/glycine levels in M-huPMCs and P-huPMCs after treatment with [13]C-glucose

*Figure 5 continued on next page*

*Figure 5 continued*

(**C**) or $^{13}$C-serine (**D**). (**E–H**) Seahorse XF analysis of M-huPMCs and P-huPMCs. Glycolytic (**E–F**) and mitochondrial functions (**G–H**) were evaluated in human Müller cells in primary culture isolated from macula and peripheral regions (n = 4).

DOI: https://doi.org/10.7554/eLife.43598.013

The following source data and figure supplement are available for figure 5:

**Source data 1.** Source data for *Figure 5C,D,F,H*.

DOI: https://doi.org/10.7554/eLife.43598.015

**Figure supplement 1.** The ratio of serine (**A**) or glycine (**B**) MS intensity to total amount of cellular protein in M-huPMCs and P-huPMCs from the same donor.

DOI: https://doi.org/10.7554/eLife.43598.014

that human Müller cells from the macula expressed less GS than those from the retinal periphery (*Daruich et al., 2018*).

We have previously reported that de novo serine synthesis plays important roles in Müller cellular redox homeostasis and mitochondrial function (*Zhang et al., 2018*). RNA-seq analysis in the present study found that macular Müller cells expressed higher levels of several key enzymes (with high RPKM numbers) involved in this metabolic pathway, such as PHGDH, than peripheral Müller cells. Our Western blot analysis confirmed that the macula expressed more PHGDH than the peripheral retina (*Figure 3*). The co-localization of PHGDH with the Müller cells marker CRALBP (*Figure 4*) is also consistent with a previous report that PHGDH is predominantly expressed by Müller cells in retina (*Zhang et al., 2018*). The strong expression of PHGDH in Henle's fibre layer, a specialised region of the macula, may explain why macular Müller cells express more PHGDH than those from the retinal periphery.

The macula has the highest metabolic activity in the retina related to the very high density of neurons, especially cone photoreceptors within the fovea (*Okawa et al., 2008*; *Perkins et al., 2004*; *Perkins and Frey, 2000*). As the major supporting glial cell in the retina, it is not surprising that human macular Müller cells display a more active glycolytic and mitochondrial metabolism than peripheral Müller cells. Our Seahorse XF experiments found that macular Müller cells had a significantly higher capacity for basal mitochondrial respiration, and ATP production, as well as higher glycolytic capacity and reserve than peripheral Müller cells (*Figure 5*). In both experiments, the readings (ECAR or OCR) started at a similar level, indicating that both populations of cells in primary culture had compatible baseline metabolism rates.

The higher expression of PHGDH in macular Müller cells is consistent with more active de novo serine synthesis in the macula than in the retinal periphery. Studies using C$^{13}$-glucose tracer revealed that Müller cells in primary cell culture had very high de novo serine/glycine enrichment (around 18% within 3 hr), much higher than those reported for melanoma cells, where only around 12% of serine was labeled with C$^{13}$-glucose within 24 hr (*Scott et al., 2011*). We also found that M-huPMCs shifted glucose carbons more toward de novo serine synthesis but less to glycolysis and the TCA cycle than P-huPMCs. Increased activity of the serine pathway in macular Müller cells is consistent with the generally higher metabolic activity in the macula.

We further explored whether the topographic variation in PHGDH expression will affect the responses of Müller cells to exogenous stimuli. Treatment with the PHGDH-specific inhibitor CBR-5884 led to more pronounced cytotoxicity in macular Müller cells than in peripheral Müller cells, after exposure to oxidative stress. This suggests that defects in de novo serine synthesis increase the vulnerability of M-huPMCs to oxidative stress relative to P-huPMCs. This observation aligns with a recent report that PHGDH dysfunction may be related to the pathogenesis of MacTel, a retinal disease characterised by pronounced degeneration of Müller cells that is confined to the macula (*Scerri et al., 2017*). This study also found serum levels of serine in MacTel patients to be significantly decreased (*Scerri et al., 2017*). Increased levels of oxidative stress, combined with PHGDH dysfunction, could thus increase the susceptibility of the macula to damage.

The greater rate of de novo serine synthesis in M-huPMCs may be linked to their increased levels of reduced GSH and ROS; this is consistent with higher metabolic activity and oxidative stress in the macula. Inhibition of de novo serine synthesis resulted in a more significant decrease in reduced GSH levels and increase in ROS levels in M-huPMCs than in P-huPMCs. This further supports the

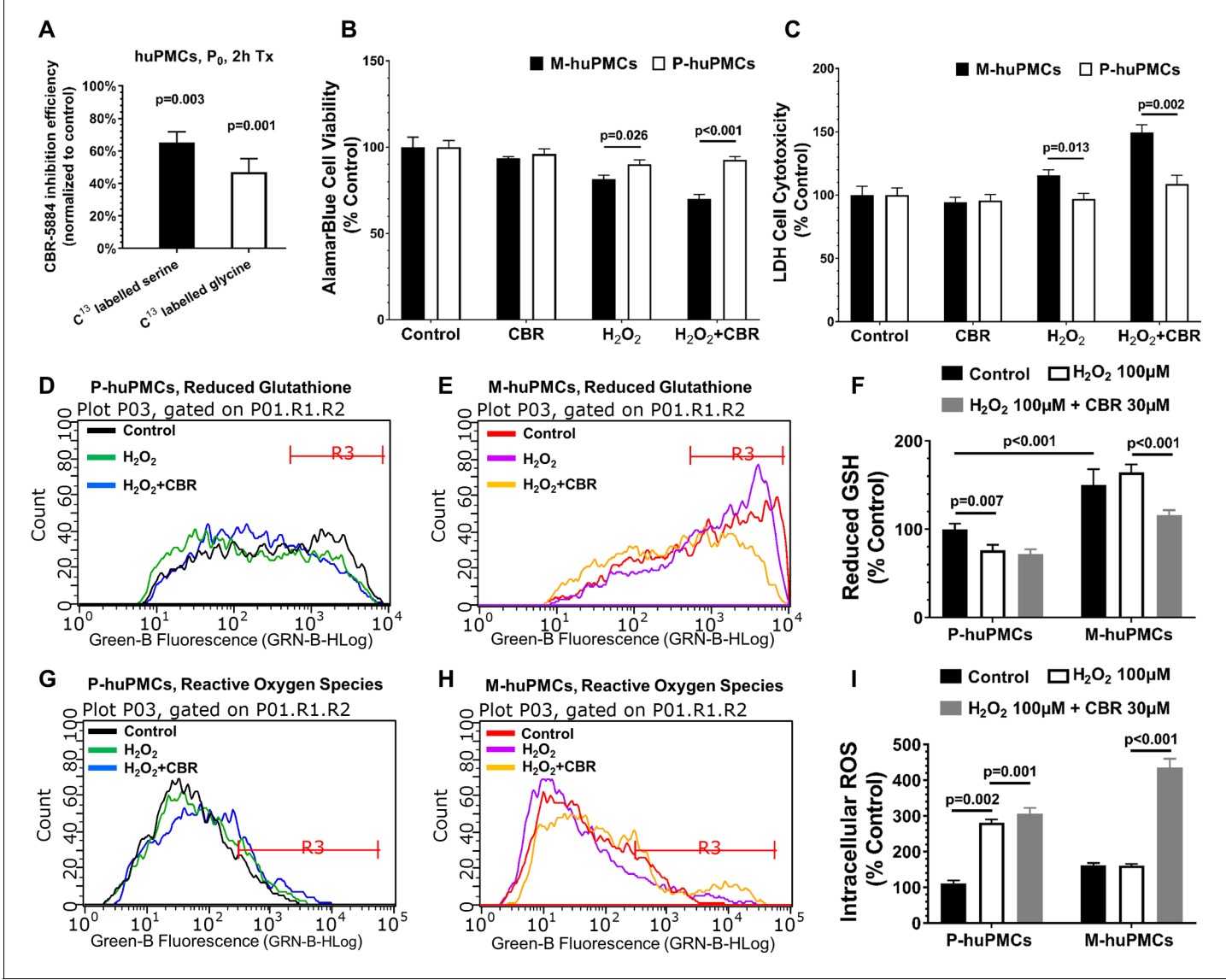

**Figure 6.** Responses to oxidative stress of primary cultured Müller cells from macula and peripheral retina. (A) Measurement of [13]C-serine and -glycine after PHGDH inhibition in human primary Müller cells; B-C. Cell metabolic activity (B) and cellular damage (C) in the M-huPMCs and P-huPMCs with or without PHGDH inhibition under oxidative stress; D-I. Flow cytometry analysis for M-huPMCs and P-huPMCs stained with Thiol Green Dye to detect GSH (D-F) and with CM-$H_2$DCFDA to evaluate ROS (G-I) for cells under mild oxidative stress (100 μM $H_2O_2$), with or without PHGDH inhibition (n = 5).
DOI: https://doi.org/10.7554/eLife.43598.016

The following source data and figure supplement are available for figure 6:

**Source data 1.** Source data for *Figure 6A–C,F,I*.
DOI: https://doi.org/10.7554/eLife.43598.018
**Figure supplement 1.** Quantitative re-analysis of the flow cytometry data in *Figure 6D–E,G–H*.
DOI: https://doi.org/10.7554/eLife.43598.017

hypothesis that impaired de novo serine synthesis can affect ROS balance in macular Müller cells more than in the peripheral retina.

These findings reveal new lines of investigation for understanding the pathogenesis of AMD and DR; the role of Müller cells in both these disease entities has received relatively little attention. The implications for MacTel, potentially a prototypic Müller cell disease (*Charbel Issa et al., 2013*), are arguably easier to draw. Although MacTel has been reported to be present in 0.1% of the population over 40 years of age, the actual prevalence may be higher (*Coorey et al., 2012*; *Klein et al.,*

2010). MacTel affects both eyes with a predilection for the temporal parafovea (*Gass and Blodi, 1993*). Recent clinicopathological studies have consistently found loss of Müller cells markers from the macula. The area of macular luteal pigment depletion, which is a specific and sensitive marker of MacTel clinically, was found to match the area of Müller cell loss, which was in turn associated with photoreceptor degeneration (*Powner et al., 2010*; *Powner et al., 2013*). It is now suspected Mac-Tel is caused by a derangement of glial-neural function with a secondary vasculopathy that is not the prime cause of loss of vision. Defects of genes associated with the PHGDH gene have recently been implicated in patients with MacTel (*Scerri et al., 2017*). Our results indicate that impairment of PHGDH would have more significant effects on macular Müller cells, especially under oxidative stress. Interestingly, we found PHGDH is highly expressed in macular Henle fibre layer, which may explain why MacTel is tightly associated with HFL pathology (*Powner et al., 2010*).

We have explored the differences between Müller cells from the macula and peripheral human retina. The distinct transcriptional and functional differences we found between these two populations of primary Müller cells contributes to explaining the unique biochemical and metabolic specializations of the human macula. We accept Müller cells in vitro, are different to Müller cells in vivo, since they have lost contact with neurons and blood vessels. Nevertheless, the Müller cells that had been isolated from the different retinal regions responded consistently and differently to the conditions they were exposed to, indicating some molecular differences between the Müller cells from these two different locations. The macula is devastatingly affected by several common blinding conditions including AMD and DR, major causes of vision impairment and loss worldwide. The differences in serine biosynthesis between Müller cells from two regions is consistent with the hypothesis that disease of Macular Müller cells causes metabolic dysfunction which is the primary cause of Mac-Tel. A better understanding of the unique biology of the human macula, both in terms of its resiliency and vulnerability, will significantly inform future approaches to disease prevention and treatments.

# Materials and methods

**Key resources table**

| Reagent type (species) or resource | Designation | Source or reference | Identifiers | Additional information |
|---|---|---|---|---|
| Antibody | Rabbit polyclonal anti-3-PGDH | Millipore | Cat# ABS571, RRID:AB_2783876 | IF(1:1000), WB (1:1000) |
| Antibody | Rabbit polyclonal anti-α/β tubulin | Cell Signaling Technology | Cat# 2148, RRID:AB_2288042 | WB (1:1000) |
| Antibody | Mouse monoclonal anti-CRALBP | Abcam | Cat# ab15051, RRID:AB_2269474 | IF(1:500) |
| Antibody | Goat polyclonal anti-GFAP | Abcam | Cat# ab53554, RRID:AB_880202 | IF(1:1000) |
| Antibody | Rabbit polyclonal anti-carbonic anhydrase II | Abcam | Cat# ab6621, RRID:AB_305602 | IF(1:1000) |
| Antibody | Rabbit polyclonal anti-Sox9 | Millipore | Cat# AB5535, RRID:AB_2239761 | IF(1:1000) |
| Recombinant DNA reagent | PHGDH-shRNA-GFP plasmid | Sigma Aldrich | MISSION SHRNA PLASMID DNA NM_006623/TRCN0000233032 / -hPGK-Neo - CMV-tGFP | |
| Recombinant DNA reagent | Negative control-shRNA-GFP plasmid | Sigma Aldrich | MISSION SHRNA CUSTOM DNA NEGATIVE CONTROL - nontarget shRNA (SHC016) - pLKO.1-Neo-CMV-tGFP vector | |
| Commercial assay or kit | Intracellular GSH detection kit | Abcam | ab112132 | |

*Continued on next page*

*Continued*

| Reagent type (species) or resource | Designation | Source or reference | Identifiers | Additional information |
|---|---|---|---|---|
| Commercial assay or kit | CM-H$_2$DCFDA | Molecular Probes | C6827 | |
| Commercial assay or kit | Pierce LDH cytotoxicity assay kit | ThermoFisher Scientific | 88953 | |
| Commercial assay or kit | AamarBlue cell viability reagent | ThermoFisher Scientific | DAL1100 | |
| Chemical compound, drug | CBR-5884 | Sigma Aldrich | SML1656 | |
| Software, algorithm | SPSS | SPSS | RRID:SCR_002865 | |

## Isolation, culture and identification of primary Müller cells from the macula and peripheral human retina

Human retinas were obtained from *post-mortem* donor eyes with ethical approval from Human Research Ethics Committee of the University of Sydney (HREC#:16/282). Human retinas without known retinal diseases were isolated as described previously (*Zhang et al., 2018*). The dissected retina was immersed in DMEM medium in a 92 mm culture dish with transparent background. The *macula lutea* was readily visualized with bright yellow macula pigment. As demonstrated in *Figure 1*, a 5 mm tissue punch centred on the central retina as well as the superior and inferior mid-periphery was taken. The mid-periphery was defined as the mid-point between the edge of the *macula lutea* and the *ora serrata*. Primary Müller cells were cultured according to our established laboratory protocol (available upon request). After retinal pieces from macula and peripheral region were cultured in DMEM medium for 6–8 weeks (with twice weekly medium change), immunofluorescent staining of Müller cell markers (GFAP, carbonic anhydrase II, SOX9 and CRALBP) was performed on the matched sets of primary Müller cells (P0, without subculturing) isolated from the macula and peripheral regions of each donor eye. Images were taken with the Olympus microscope (IX71).

## RNA sequencing

After extracting the total RNA from M-huPMCs and P-huPMCs (n = 8), mRNA was enriched using the oligo (dT) magnetic beads. The library preparation, sequencing and quality control were commercially contracted to BGI (https://www.bgi.com/global/). The mRNA was fragmented into short fragments (200 ~ 700 bp) in the fragmentation buffer. The first-strand cDNA was synthesized with random hexamer-primer using the mRNA fragments as templates, followed by the second strand synthesis. The double stranded cDNA was purified with a QiaQuick PCR extraction kit and then used for end repair and base A addition. Finally, sequencing adapters were ligated to the fragments. The fragments were purified by SPRI bead size selection and enriched by PCR amplification. The library products were sequenced using Illumina HiSeq 2500 with paired end 100 bp read length.

## RNA data analysis

Primary sequencing data was generated by Illumina HiSeq 2500. Raw reads were filtered to remove adaptor sequences and PCR duplicates. The filtered clean reads were aligned to the reference sequences with SOAP2. The alignment data was utilized to calculate distribution of reads on reference genes and perform coverage analysis. Downstream analysis was performed including gene differential expression analysis (DESeq v1.18.0), alternative splicing (tophat v2.0.8+cufflinks v2.0.2) and SNP detection (using SOAPsnp v1.05). Results of gene expression included gene expression levels and differential expression analysis was performed using DESeq2 in R (version 3.5.1). P-values were adjusted for multiple testing using the Benjamini-Hochberg procedure (*Benjamini and Hochberg, 1995*). A false discovery rate (FDR) adjusted p-value (i.e. q-value)<0.05 was set for the selection of differential expression genes. Two dimensional plots of principal components were calculated with principal component analysis using R software. We used ClusterProfiler (Bioconductor; https://

bioconductor.org/pack-ages/release/bioc/html/clusterProfiler.html) which is an R package to analyse gene clusters and classify biological terms.

## Seahorse XF analysis

Macular and peripheral Müller cells were seeded at a density of $2 \times 10^4$ cells/well in DMEM medium into the Seahorse XF96 cell culture microplates (Seahorse Bioscience, Agilent Technologies, Santa Clara, CA, USA). The mitochondrial stress assay was carried out in assay medium containing XF base medium (Aligent). Assay medium was freshly prepared and adjusted to pH 7.4. After 24 hr incubation at 37°C in 5% $CO_2$, the confluent cells were ready to be tested. After programming in accordance with manufacturer's recommendation, the testing plate was placed in Seahorse XF Analyser. Oxygen consumption rates (OCR) and extracellular acidification rates (ECAR) were measured approximately every 8 min. The stress reaction system in each well contained 175 µl assay buffer, 25 µl Oligomycin, 25 µl FCCP and 25 µl Rotenone/Antimycin A to achieve the working concentrations of 1.26 µM, 1 µM, 0.5 µM respectively, and the drugs were sequentially added to each well at different time points according to manufacturer instructions. The OCR and ECAR curves were obtained as the reads of stress resulted from drugs at different time points. The rate is normalized to the amount of protein in each well. The differences in glycolytic and mitochondrial functions were compared between the two primary Müller cell populations (macula and peripheral retina).

## LDH cell cytotoxicity assay and AlamarBlue cell viability assay

M-huPMCs and P-huPMCs (P0) were detached from culture dishes using TrypLE enzyme (Thermo Fisher Scientific) and seeded at a density of $1 \times 10^4$ cells/well in 96-well plates. Cells were cultured to 80% confluence in DMEM/10% FBS and then starved overnight with Basal Medium Eagle (Gibco) supplemented with 1% FBS, 1% penicillin-streptomycin (P/S) and 1% Insulin-Transferrin-Selenium (ITS, Thermo Fisher Scientific). M-huPMCs and P-huPMCs (P1) were then treated with 100 µM $H_2O_2$, 15 µM CBR-5884 and 100 µM $H_2O_2$ + 15 µM CBR-5884 for 6 hr, respectively. A medium only, no treatment group was included as a control. The LDH concentration in each group was assayed by LDH cytotoxicity assay kit (Pierce). In brief, 20 µl supernatant of the 96-well plate was transferred into a 386-well plate. 20 µl reaction mixture was added into each well of the plate. The mixture was incubated at room temperature for 30 min. 20 µl stop solution was added to cease the reaction. The absorbance was measured at 490 nm and 680 nm with the plate reader (Safire², Tecan). To assess cell viability, 15 µl alamarBlue reagent was added to 150 µl medium in each well of the 96-well plate. After a 2 hr incubation at 37°C, the absorbance was read at 570 and 600 nm with the Tecan Safire plate reader (TECAN).

## Western blot

The cells were washed with PBS and lysed in RIPA buffer (Sigma Aldrich) with Protease/Phosphatase Inhibitor Cocktail (Cell Signalling). The lysed cells were incubated at 4°C for 10 min and then centrifuged at 12,000 g at 4°C for 10 min. The supernatants were collected and protein concentrations were determined using a BCA protein assay kit (Thermo Fisher Scientific). Samples were heated with reducing buffer and NuPAGE 4–12% Bis-Tris Protein Gels (Life Technologies) at 70°C for 10 min. The samples were loaded onto NuPAGE 4–12% Bis-Tris Protein gels (Thermo Fisher Scientific) and electrophoresed at 180V, 4°C for 70 min. The proteins were transferred using the iBlot gel transfer device (Thermo Fisher Scientific). The PVDF membranes (Millipore, 0.42 uM) were blocked with 5% bovine serum albumin (BSA), Tris-Buffered Saline and 0.1% Tween 20 (TBST) for 1 hr at room temperature (RT) and then incubated with either primary antibody overnight at 4°C: PHGDH (Millipore, #ABS571, 1:1000) or α/β tubulin (Cell Signalling, #2148, 1:1000). The membranes were washed three times with PBST and incubated with the HRP-conjugated secondary antibody (1:5000) at RT for 2 hr on the following day. Then the membranes were washed with TBST for 5 min, three times and TBS for 5 min once. After washing, the membranes were incubated in Clarity Western ECL Substrate for 5 min and imaged using the G-Box imaging system (In Vitro Technologies).

## Vibratome sectioning

Retinal punches from the macula and peripheral regions of donor eyes were fixed in 4% paraformaldehyde (PFA) in PBS for 2 hr and then transferred to PBS for 1 hr. Retinal pieces were embedded in

low melting point agarose (42°C, 3% agarose (Lonza) in PBS). Tissue blocks were cooled down at 4°C until the agarose solidified. Agarose blocks were trimmed and glued to the metal chuck of a vibratome (Leica, VT1200S). Vibratome sections (100 μm thick) were obtained from retinal tissues and stored in PBS at 4°C.

## Immunofluorescent staining

HuPMCs in 24-well plates were fixed in 4% PFA for 1 hr, then rinsed and stored in PBS for later immunofluorescence staining. Retinal vibratome sections and huPMCs were initially blocked in 5% normal goat serum (Sigma) overnight at 4°C. Sections were immunostained with primary antibodies: PHGDH (Millipore, ABS571, 1:1000), CRALBP (Abcam, ab15051, 1:500), GFAP (Abcam, ab53554, 1:1000), Carbonic Anhydrase II (Abcam, ab6621, 1:1000), SOX9 (Millipore, ab5535, 1:1000) diluted in PBS containing 1% normal goat serum and 0.5% Triton X-100. Vibratome sections were incubated for 4 nights at 4°C. Thereafter, huPMCs were incubated with species-specific secondary antibodies conjugated with Alexa Fluor 488 (green) or 594 (red) (Molecular Probes) at a 1:1000 dilution for 2 hr at room temperature; while the vibratome sections were incubated at 4°C overnight. Cells and sections were then rinsed with PBS and nuclei were stained with Hoechst 33342 (Thermo Fisher Scientific). After mounting in VECTASHIELD antifade mounting medium (Vector Laboratories), immunofluorescence labelling of huPMCs and vibratome sections was captured with the ZEISS confocal laser-scanning microscope (LSM700, Carl Zeiss) and ZEN Blue software.

## $^{13}C$ Glucose labeling experiments

M-huPMCs and P-huPMCs (P0) in T25 flasks (Corning) were starved overnight with BME supplemented with 1%FBS, 1% penicillin-streptomycin, and 1% Insulin-Transferrin-Selenium (ITS). M-huPMCs and P-huPMCs were treated with 30 μM CBR-5884 for 4 hr. Cells were incubated with $^{13}C$-glucose (5 mM, CLM-1396–0.5, Cambridge Isotope Lab) with or without CBR-5884 (30 μM) in KRB (Krebs-Ringer-Bicarbonate buffer) for 2 hr in a $CO_2$ incubator. Before the incubation, KRB containing the $^{13}C$-glucoce was pre-warmed at 37°C in a $CO_2$ incubator for 2 hr. 1 ml cold 0.9% sodium chloride was used to rinse the cells, followed by 800 μl extraction buffer (methanol: dd$H_2O$, 800:200). Cells were then scraped off the flask on dry ice and transferred into Eppendorf tubes. The cell/extraction buffer mix was homogenized and placed on dry ice for 30 min, then centrifuged at 13,000 g for 10 min at 4°C, and the supernatant transferred into the clean tubes. 0.1 M NaOH was added into the pellet of and incubate at 37°C for 16 hr. Protein concentration was measured by BCA method (Thermo Fisher Scientific) (*Du et al., 2015*; *Zhu et al., 2018*).

## Flow cytometry

We detected the intracellular ROS and GSH in M-huPMCs and P-huPMCs (P0) after different treatments, using CM-$H_2$DCFDA (Molecular Probes) and Intracellular GSH Detection Kit (Abcam). The huPMCs were washed with PBS to terminate the treatment and TrypLE enzyme (Thermo Fisher Scientific) was used to detach cells from culture dishes. After centrifugation (340 g), cells from different treatments were separated into two groups and incubated with either 2 μM CM-$H_2$DCFDA or 1 × Thiol Green Dye for 1 hr in HBSS at 5% CO2, 37°C, respectively. Cells were centrifuged at 340 g for 5 min, washed with PBS and loaded onto a Guava easycyte flow cytometer (Merck Millipore). Cells were gated for singlets (GRN-B-Alog vs. GRN-B-Hlog). The fluorescent event counts were recorded, and the data was analysed using Guava easycyte software 3.1.

## Statistical analysis

The data are expressed as the means ± SEM. Statistical analyses were performed using the SPSS version 17.0 for Windows software. The differences between mean values were evaluated using one-way ANOVA, followed by Tukey post hoc test. Student's t tests were used to compare differences between any given two groups throughout the study. A p-value less than 0.05 was considered to indicate a statistically significant difference.

## Acknowledgements

We acknowledge the Lions NSW Eye Bank and Australian Ocular Biobank tissue co-ordinators and scientistsand eye donors and their families for the human donor eye tissue used in this project. The Bosch Institute Molecular Biology Facility provided support and training for the techniques and equipment used in this research. This study was supported by an Australian National Health and Medical Research Council project grant (APP1145121), The Ophthalmic Research Institute of Australia, a grant from the Lowy Medical Research Institute and NIH Grant EY026030 (JD). Professor Mark C Gillies is a Sydney Medical School Fellow and supported by an Australian National Health and Medical Research Council Practitioner Fellowship. Dr. Ling Zhu is supported by The Claffy Foundation, Sydney Hospital/Sydney Eye Hospital.

## Additional information

### Funding

| Funder | Grant reference number | Author |
|---|---|---|
| Australian National Health and Medical Research Council | APP1145121 | Ling Zhu<br>Weiyong Shen<br>Mark C Gillies |
| The Ophthalmic Research Institute of Australia | | Ling Zhu<br>Mark C Gillies |
| National Institutes of Health | EY026030 | Jianhai Du |
| Lowy Medical Research Institute | | Ting Zhang<br>Ling Zhu<br>Weiyong Shen<br>Mark C Gillies |

The funders had no role in study design, data collection and interpretation, or the decision to submit the work for publication.

### Author contributions

Ting Zhang, Conceptualization, Data curation, Software, Formal analysis, Writing—original draft; Ling Zhu, Conceptualization, Data curation, Supervision, Funding acquisition, Project administration, Writing—review and editing; Michele C Madigan, Svetlana Cherepanoff, Fanfan Zhou, Conceptualization, Writing—review and editing; Wei Liu, Shaoxue Zeng, Data curation; Weiyong Shen, Conceptualization; Jianhai Du, Mark C Gillies, Conceptualization, Resources, Data curation, Supervision, Funding acquisition, Project administration, Writing—review and editing

### Author ORCIDs

Ting Zhang (iD) https://orcid.org/0000-0001-8074-8999
Ling Zhu (iD) https://orcid.org/0000-0003-0776-1630
Michele C Madigan (iD) https://orcid.org/0000-0003-1053-6979
Shaoxue Zeng (iD) https://orcid.org/0000-0003-3597-6014

### Ethics

Human subjects: Human retinas were obtained from post-mortem donor eyes with ethical approval from Human Research Ethics Committee of the University of Sydney (HREC#:16/282).

### Decision letter and Author response

Decision letter https://doi.org/10.7554/eLife.43598.025
Author response https://doi.org/10.7554/eLife.43598.026

## Additional files

### Supplementary files

• Supplementary file 1. RNA sequencing raw data.
DOI: https://doi.org/10.7554/eLife.43598.019

• Supplementary file 2. RNA sequencing filtered data.
DOI: https://doi.org/10.7554/eLife.43598.020

### Data availability

RNA sequencing data are included in the manuscript and Supplementary files. These data are also available at Dryad (https://doi.org/10.5061/dryad.hp60p89).

The following dataset was generated:

| Author(s) | Year | Dataset title | Dataset URL | Database and Identifier |
|---|---|---|---|---|
| Zhang T, Zhu L, Madigan M, Liu W, Shen W, Cherepanoff S, Zhou F, Du J, Gillies M | 2018 | Data from: Human macular Müller cells rely more on serine biosynthesis to combat oxidative stress than those from the periphery | https://doi.org/10.5061/dryad.hp60p89 | Dryad Digital Repository, 10.5061/dryad.hp60p89 |

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
