## [Decision Letter]

Thank you for submitting your article "Human macular Müller cells rely more on serine biosynthesis to combat oxidative stress than those from the periphery" for consideration by *eLife*. Your article has been reviewed by Gary Westbrook as the Senior Editor, a Reviewing Editor, and three reviewers. The following individuals involved in review of your submission have agreed to reveal their identity: James B Hurley (Reviewer #1); Paul Bernstein (Reviewer #2). The reviewers have discussed the reviews with one another and the Reviewing Editor has drafted this decision to help you prepare a revised submission.

Summary:

Three experts reviewed your manuscript, and their assessments, together with my own, form the basis of this letter. As you will see, all of the reviewers were impressed with the importance and novelty of your work. I am including the three reviews at the end of this letter, as there are a variety of specific and useful suggestions in them. One that we think is of particular importance (noted by reviewers #1 and #3) is the quantification on the immunoblot in Figure 3. We appreciate that the reviewers' comments cover a broad range of suggestions for improving the manuscript. Please use your best judgment in deciding which of these can be accommodated in a reasonable period of time. We look forward to receiving your revised manuscript

*Reviewer #1:*

This manuscript characterizes differences between cultured primary Müller cells isolated from either the macula or from the periphery of human retinas. The study acknowledges and addresses limitations of cultured Müller cells as representative of bona fide Müller cells in the retina of an eye. The study finds many structural and functional differences. Much of the paper focuses on a quantitative difference in serine metabolism. The report, with its acknowledged limitations, is an important contribution that will help to advance the understanding of degenerative diseases that preferentially affect the macula.

The authors should address the following concerns and suggestions.

1) Figure 1B shows the "Cytoplasma Neucleus" ratio, but the text in subsection “Isolation, culturing and validation of primary Müller cells from human macula and 95 peripheral retina” refers to the nucleus/cytoplasm ratio. Please make these consistent.

2) Subsection “Isolation, culturing and validation of primary Müller cells from human macula and peripheral retina”. It would be useful to include other well-known markers for Müller cells, like CRALBP and Glutamine Synthetase. It should be noted or commented on that GFAP expression increases when Müller cells are stressed.

3) The quantification of the western blot data shown in Figure 3B is very useful. However, in the raw data shown in Figure 3A it looks like there is a decrease in the amount of tubulin that is perhaps even greater than the increase in the amount of PHGDH. Since the data in Figure 3B is expressed as a ratio I think there needs to be a more convincing way to show that PHGDH levels are different when normalized to total protein or to another marker. It is possible that the tubulin content in the macular vs peripheral Müller cells is different. That would be significant and should be discussed.

4) Subsection “Differential expression of genes related to the de novo serine synthesis pathway in Müller cells isolated from the human macula and peripheral retina” – the reference to Zhange et al., 2018 is confusing here – is that cited as an example of inter-individual variations? Please clarify what this citation is supporting.

5) Please include more information about how the various antibodies were validated. This is especially important for Figures like Figure 4. Has this immunoreactivity been confirmed to be negative against knockouts? Does the staining pattern depend on the concentration of antibody? How did the authors decide what antibody concentration to use? Was a titration done?

6) For Figure 5 please report the total amounts of serine and glycine per mg tissue as this might also change (in addition to the changes in% enrichment).

7) Subsection “Metabolic differences between the macular and peripheral Müller cells”: This is a confusing statement because the level of 13C labeled M2 glycine seems to be only slightly higher. (~70% vs ~60%) in the macular cells than in the peripheral cells.

8) The data in Figure 5A-D might be easier to interpret if there was more than one time point. It is hard to know whether these numbers represent kinetic differences or if they represent differences in the sizes of the pools that are accessible to the added labeled glucose or labeled serine.

9) For the oximetry experiments in Figure 5E-FH, how were the rates normalized – was it per number of cells, per mg of protein? Please describe this either in the Figure itself or in the Figure legend.

10) Figure 6A. How was the optimal concentration of CBR-5884 determined? In Figure 6A it appears that it is not very effective. Please comment on this. Also, does it affect the total amount of serine and glycine or just the% that is labeled? Can higher concentrations be used and is it known if it affects other reactions?

11) Please interpret the flow cytometry data. The Macular-derived cells show a stronger signal in Figure 6E but is that simply because they have a different size or morphology? Is there a control that can be used to dismiss that explanation?

12) In the Figure legend for Figure 6 please state the methods used in flow cytometry to detect GSH and ROS.

13) There are obvious concerns about the relevance of differences between cells that are isolated from other cells, proteolyzed and exposed to culture conditions. A strength of this paper is that the authors acknowledge and address these concerns. They discuss the caveats and point out how this study is useful and relevant in the Discussion section of the manuscript. (no response required for this point).

14) Please cite a reference or data that support the statement in the Discussion section: "The macula has the highest metabolic activity in the retina related to the very high density of neurons, especially cone photoreceptors within the fovea."

15) One way to provide further validation of the cultured Müller cells would be to measure the rate of 13C Serine synthesis from glucose in intact retina and compare it to the rate in the cultured cells. If there is a way to normalize the samples so that they contain approximately the same number of Müller cells then those two types of preparation should also make about the same amount of 13C serine. If the cultured cells make more serine then there would be a problem. But if they make about the same, then the validity of the preparation is supported. If this is a feasible experiment it would strengthen the paper.

16) Please correct the grammar in the sentence “We further explored whether the topographic variation in PHGDH expression of 348 affected the responses of Müller cells to exogenous stimuli.”. (Discussion section)

17) Subsection “13C pulse‐chase labeling Experiments”: The subtitle here reports that it is a "pulse-chase" experiment, but it seems more like the 13C glucose label is present throughout the experiment. If that is correct it should not be referred to as a "pulse-chase".

18) Subsection “13C pulse‐chase labeling Experiments”: "glucoce"

*Reviewer #2:*

This manuscript by Zhang and colleagues comprehensively and rigorously addresses the question of whether or not Muller cells of the macula substantially differ from their compatriots in the periphery with particular attention to serine metabolism and its role in combating oxidative stress. This is an important question because the macula is selectively affected by a number of blinding conditions, and although Muller cells are present throughout the retina, there is accumulating morphological and biochemical evidence that macular Muller cells differ from their counterparts in the periphery. Additionally, we are now beginning to recognize that in diseases such as MacTel, the Muller cells are intimately involved in the underlying pathogenesis. Since MacTel has been linked to abnormalities of serine metabolism and PHGDH defects, their focus on this particular pathway and gene product makes sense.

Overall, the logic and execution of the presented work are excellent. They first show that they can culture the macular and peripheral Muller cells from human donors successfully and that their primary culture cells express the expected markers and show markedly different morphologies depending on their region of origin. Gene expression studies confirm these findings and show that PHGDH is relatively upregulated in the macula, especially in the fovea. They then go on to show biochemically that serine-related pathways may contribute resistance to oxidative stress in the macula.

All in all, at the conclusion of reading this manuscript, I felt convinced that the authors have demonstrated that macular Muller cells are unique and that they have plausible explanations that can help elucidate the underlying cause of maculopathies such as MacTel. This is well done and important work.

*Reviewer #3:*

This manuscript compares human primary Muller cell cultures from macular and peripheral retina in morphology, transcriptome, serine biosynthesis, metabolism, and response to oxidative stress. The author argues that macular Muller cells have an augmented de novo serine synthesis system with higher expression levels of key enzymes. This is necessary for macular Muller cells to handle graver oxidative stress associated with higher metabolic rates in the macular region. Overall, it is an interesting and important subject pertaining to common human retinal diseases such as AMD and diabetic retinopathy, and it also has a broader impact regarding the role of glial metabolism in the health of neural tissues in general.

One obvious caveat is that Muller cells in vitro may be different from those in retinal tissues. Especially the process of generating Muller cell cultures is a form of stress that kills off all neurons and other cells. However, for human/primate cells, other than single cell sequencing (such as the work in https://doi.org/10.1101/428110), and potential studies using retinal organoids, this primary culture system is a reasonably good alternative for a relatively pure Muller cell population study.

My major concerns are mostly technical in nature. Overall, there is a lack of necessary details in methods that casts doubt on some of the results.

On the sequencing part, the authors neglected to report important details: what is the read length? single or paired end reads?

Subsection “Transcription profiles of macular and peripheral Müller cells”, "A total of 21,338 differentially expressed genes (DEGs) were identified in Müller cells isolated from human macular and peripheral retina." This number is quite high, nearly as high as the total number of annotated genes in humans.

Subsection “Transcription profiles of macular and peripheral Müller cells”, "14,380 DEGs with a q-value (a p-value that has been adjusted for the False Discovery Rate) < 0.05 were detected". So about 2/3 of all human genes are differentially expressed in these two populations?

On Western blot, the loading controls are problematic. In Figure 3A (western blots on cultured cells) α/β tubulin was used as loading control. However, from their sequencing data, TUBB is 1.37x higher in M-huPMC, but TUBA1A is 1.64x lower. In the rest of Figure 3 (western blots from retinal tissues), tubulin was again used as loading control. However, it is clear that the tubulin band varies significantly across different samples, with an overall tendency to be higher in the macular regions. If tubulin and PHGDH vary independently across retinal regions, it cannot be a reasonably good loading control, thus the result in Figure 3D becomes somewhat questionable. Unfortunately, this is the critical data connecting cultured results with retinal tissues.

On mitochondria respiration measurements, judging from baseline data points before oligomycin in Figure 5G, it is difficult to relate it to the "basal respiration" data in Figure 5H, which shows a significant difference. Overall, the ECAR data looks much more convincing than the OCR data.

[Editors’ note: minor comments from Reviewers 2 and 3 have not been included.]

---

## [Author Response]

Summary:Three experts reviewed your manuscript, and their assessments, together with my own, form the basis of this letter. As you will see, all of the reviewers were impressed with the importance and novelty of your work. I am including the three reviews at the end of this letter, as there are a variety of specific and useful suggestions in them. One that we think is of particular importance (noted by reviewers #1 and #3) is the quantification on the immunoblot in Figure 3. We appreciate that the reviewers' comments cover a broad range of suggestions for improving the manuscript. Please use your best judgment in deciding which of these can be accommodated in a reasonable period of time. We look forward to receiving your revised manuscriptReviewer #1:This manuscript characterizes differences between cultured primary Müller cells isolated from either the macula or from the periphery of human retinas. The study acknowledges and addresses limitations of cultured Müller cells as representative of bona fide Müller cells in the retina of an eye. The study finds many structural and functional differences. Much of the paper focuses on a quantitative difference in serine metabolism. The report, with its acknowledged limitations, is an important contribution that will help to advance the understanding of degenerative diseases that preferentially affect the macula.The authors should address the following concerns and suggestions.1) Figure 1B shows the "Cytoplasma Neucleus" ratio, but the text in subsection “Isolation, culturing and validation of primary Müller cells from human macula and 95 peripheral retina” refers to the nucleus/cytoplasm ratio. Please make these consistent.

We have revised the text to cytoplasm/nucleus ratio. (subsection “Isolation, culturing and validation of primary Müller cells from human macula and - peripheral retina”) (Discussion section.)

2) Subsection “Isolation, culturing and validation of primary Müller cells from human macula and peripheral retina”. It would be useful to include other well-known markers for Müller cells, like CRALBP and Glutamine Synthetase. It should be noted or commented on that GFAP expression increases when Müller cells are stressed.

Several representative Müller cells markers, including CRALBP and glutamine synthetase (GS), were analysed in our RNA sequence results (Table 1). We have also performed immunostaining for CRALBP in a matched pair of huPMCs from the macula and peripheral retina as shown below (Figure 1—figure supplement 1). We have also included a sentence related to Müller cells and GFAP in the revised manuscript as follows “GFAP is a non-specific response of Müller cell stress which they usually express in cell culture (Augustine et al., 2018).” (subsection “Isolation, culturing and validation of primary Müller cells from human macula and - peripheral retina”).

3) The quantification of the western blot data shown in Figure 3B is very useful. However, in the raw data shown in Figure 3A it looks like there is a decrease in the amount of tubulin that is perhaps even greater than the increase in the amount of PHGDH. Since the data in Figure 3B is expressed as a ratio I think there needs to be a more convincing way to show that PHGDH levels are different when normalized to total protein or to another marker. It is possible that the tubulin content in the macular vs peripheral Müller cells is different. That would be significant and should be discussed.

We did not state clearly in the figure legend that Figure 3A is an immunoblot for Donor 1 only, there were three technical repeats, as a representative image for this experiment. Figure 3B is the quantitative analysis of the Western blots for all four donors. We chose an antibody anti-α/β-Tubulin (Cell signalling, #2148) as the loading control to be consistent with our western blot of retinal total protein in Figure 3F since it detects endogenous levels of both α- and β-tubulin total protein. We have now provided the western blots of all four sets of human primary Müller cells from four donors using β-actin as the loading control, with similar findings shown in the Figure 3A and an updated quantitative analysis based on this blot (Figure 3B).

4) Subsection “Differential expression of genes related to the de novo serine synthesis pathway in Müller cells isolated from the human macular and peripheral retina” – the reference to Zhange et al., 2018 is confusing here – is that cited as an example of inter-individual variations? Please clarify what this citation is supporting.

Thank you. The reference was a miss insert and this has been revised in the manuscript.

5) Please include more information about how the various antibodies were validated. This is especially important for Figures like Figure 4. Has this immunoreactivity been confirmed to be negative against knockouts? Does the staining pattern depend on the concentration of antibody? How did the authors decide what antibody concentration to use? Was a titration done?

Most antibodies have already been validated by many reports (CRALBP (Yu et al., 2010), GFAP (Sarfare et al., 2015); Carbonic Anhydrase II (Cui et al., 2012); *SOX9* (Muranishi and Furukawa, 2012)). We have now transfected PHGDH-shRNA-GFP (TRCN0000233032: NM_006623.2-1563s21c1) plasmid into the MIO-M1 Müller cell Line to knockdown PHGDH expression to assess PHGDH immunolabelling. We performed immunostaining three days after the transfection using anti-PHGDH antibody (1:1000) on fixed cells (Figure 4—figure supplement 1). Cells expressing PHGDH-shRNA-GFP (green fluorescence, white arrow, D) expressed less PHGDH (red fluorescence, white arrow, E). In comparison, there was normal PHGDH immunoreactivity (red fluorescence, yellow arrow, B) observed in cells transfected with negative control shRNA plasmid (Scramble sequence, green fluorescence, yellow arrow, A). This finding is consistent with the anti-PHGDH antibody we used specifically recognise human PHGDH protein.

We used serial dilution of antibodies to determine the optimal antibody concentration to be used in the immunostaining.

6) For Figure 5 please report the total amounts of serine and glycine per mg tissue as this might also change (in addition to the changes in% enrichment).

Total serine MS intensity divided by total amount of cellular protein was used to compare the relative levels of serine or glycine in macular and peripheral-huPMCs isolated from the same donor retina. We believe that this ratio should detect any difference in serine or glycine levels between Müller cells from the 2 different regions (Figure 5—figure supplement 1). We did not observe a significant difference in the ratio of serine (A) or glycine (B) to total protein in M-huPMCs and P-huPMCs.

7) Subsection “Metabolic differences between the macular and peripheral Müller cells”: This is a confusing statement because the level of 13C labeled M2 glycine seems to be only slightly higher. (~70% vs ~60%) in the macular cells than in the peripheral cells.

We thank the reviewer for the comment. We have revised the manuscript as follows: “A modest but statistically significantly higher level of ^13^C-labeled M2 glycine (70% vs 59%) suggests that more serine is converted to glycine in M-huPMCs than in P-huPMCs.” (Subsection “Metabolic differences between the macular and peripheral Müller cells”).

8) The data in Figure 5A-D might be easier to interpret if there was more than one time point. It is hard to know whether these numbers represent kinetic differences or if they represent differences in the sizes of the pools that are accessible to the added labeled glucose or labeled serine.

We strictly used only P0 human primary Müller cells in this experiment. The number of primary Müller cells isolated from each macula is very limited and we must use the macular and peripheral cells from the same donor to compare, so technically we were unable to test more time points with the limited amount of P0 cells with current protocols.

9) For the oximetry experiments in Figure 5E-FH, how were the rates normalized – was it per number of cells, per mg of protein? Please describe this either in the Figure itself or in the Figure legend.

The rate is normalized to the amount of protein in each well. We have revised the Materials and methods section to indicate this accordingly. (Subsection “Seahorse XF Analysis”).

10) Figure 6A. How was the optimal concentration of CBR-5884 determined? In Figure 6A it appears that it is not very effective. Please comment on this. Also, does it affect the total amount of serine and glycine or just the% that is labeled? Can higher concentrations be used and is it known if it affects other reactions?

We selected a concentration that we thought would be effective without reducing cellular metabolic activity based on a report from Mullarky et al., (2016). We are not aware of off-target effects of CBR-5884, but we still minimized the treatment duration to avoid any possible irrelevant off-target effects. CBR-5884 was reported to decrease de novo serine synthesis by 30% (changes of ion intensity, ^13^C-glucose labelling experiment, (Mullarky et al., 2016)), which is consistent with our findings in human primary Müller cells.

11) Please interpret the flow cytometry data. The Macular-derived cells show a stronger signal in Figure 6E but is that simply because they have a different size or morphology? Is there a control that can be used to dismiss that explanation?

Forward Scatter (FSC) and Side Scatter (SSC) were used to identify cells based on size and granularity, which are reflections of complexity. Both M-huPMCs and P-huPMCs had a similar profile of FSC/SSC distribution after trypsinization, suggesting that they were of a similar size and complexity. We used the same gating for both FSC and SSC to exclude cells of different size or complexity in the sample cohorts. We also re-analysed the different responses of M-huPMCs and P-huPMCs to induced stress. The extent of the increase of ROS and decrease of GSH in treated M-huPMCs and P-huPMCs has been normalized to their own control (Figure 6—figure supplement 1).

12) In the Figure legend for Figure 6 please state the methods used in flow cytometry to detect GSH and ROS.

We have now provided details in the figure legend: “Flow cytometry analysis for M-huPMCs and P-huPMCs stained with Thiol Green Dye to detect GSH (D-F) and with CM-H_2_DCFDA to evaluate ROS (G-I) for cells under mild oxidative stress (100 μM H2O2), with or without PHGDH inhibition (n=5).”

14) Please cite a reference or data that support the statement in the Discussion section: "The macula has the highest metabolic activity in the retina related to the very high density of neurons, especially cone photoreceptors within the fovea."

We have now included references (Okawa et al., 2008; Perkins et al., 2004; Perkins and Frey, 2000) as suggested (Discussion section). Cone photoreceptors have been reported to have higher energy demands than rod photoreceptors. In view of the highest density of cones, it may be speculated that the macula has the highest metabolic activity in the retina.

15) One way to provide further validation of the cultured Müller cells would be to measure the rate of 13C Serine synthesis from glucose in intact retina and compare it to the rate in the cultured cells. If there is a way to normalize the samples so that they contain approximately the same number of Müller cells then those two types of preparation should also make about the same amount of 13C serine. If the cultured cells make more serine then there would be a problem. But if they make about the same, then the validity of the preparation is supported. If this is a feasible experiment it would strengthen the paper.

We have now established a method to measure the rate of ^13^C Serine synthesis from glucose and performed the experiments on four donor pairs of post-mortem human macula and mid-peripheral retina (5mm diameter punches) as suggested by the reviewer. We found that 3 out of 4 macula explants had higher rates of serine metabolism when compared with peripheral retina from the same donor. (Author response image 1).

Due to the limited sample size, statistical analysis was not possible. The post-mortem delay, age of donor and the cause of death may also contribute to the variations between the different donor retinas. Further analyses of larger numbers would be helpful.

Another challenge is how much serine in each cell and which cell uses de novo synthesis. For example, if there is more serine in other cells that can dilute the rate of enrichment. Also, the dead cells in culture can generate serine by degrading protein, further complicating this analysis. We also need to adopt a method of counting the density of Müller cells within the explants to estimate the number of Müller cells in macula or mid-peripheral region. One of our colleagues’ team is now counting the different types of retinal cells in different area of human retina however, this will be a separate publication and beyond the scope of the current study.

**Author response image 1. respfig1:** ^13^C-labelled serine level in the macula and peripheral retina following the treatment of ^13^C-glucose (n=4 donors). Donors 1, 2 and 3 showed a higher ^13^C-labelled serine level in the macula *vs* peripheral retina. Donor 4 showed reduced ^13^C-labelled serine in the macula.

16) Please correct the grammar in the sentence “We further explored whether the topographic variation in PHGDH expression of 348 affected the responses of Müller cells to exogenous stimuli.”. (Discussion section)

Thank you. This has been revised as suggested (Discussion section).

17) Subsection “13C pulse‐chase labeling Experiments”: The subtitle here reports that it is a "pulse-chase" experiment, but it seems more like the 13C glucose label is present throughout the experiment. If that is correct it should not be referred to as a "pulse-chase".

Thank you. We have revised this accordingly “^13^C Glucose Labeling Experiments”.

18) Subsection “13C pulse‐chase labeling Experiments”: "glucoce"

Thank you. We have revised accordingly.

Reviewer #3:

[…] My major concerns are mostly technical in nature. Overall, there is a lack of necessary details in methods that casts doubt on some of the results.On the sequencing part, the authors neglected to report important details: what is the read length? single or paired end reads?Subsection “Transcription profiles of macular and peripheral Müller cells”, "A total of 21,338 differentially expressed genes (DEGs) were identified in Müller cells isolated from human macular and peripheral retina." This number is quite high, nearly as high as the total number of annotated genes in humans.Subsection “Transcription profiles of macular and peripheral Müller cells”, "14,380 DEGs with a q-value (a p-value that has been adjusted for the False Discovery Rate) < 0.05 were detected". So about 2/3 of all human genes are differentially expressed in these two populations?

We apologize for not applying the appropriate cut-off for the number of DEGs here. The correct number with the proper cut-off applied is now shown in Figure 2F. The correction was made as follows: " A total of 7,588 differentially expressed genes (DEGs) were identified in Müller cells isolated from human macular and peripheral retina with a 1.5-fold or more increase or decrease and with an FDR corrected p-value < 0.05.” (Figure 2F, Supplementary file 1 and Supplementary file 2 updated).

On Western blot, the loading controls are problematic. In Figure 3A (western blots on cultured cells) α/β tubulin was used as loading control. However, from their sequencing data, TUBB is 1.37x higher in M-huPMC, but TUBA1A is 1.64x lower. In the rest of Figure 3 (western blots from retinal tissues), tubulin was again used as loading control. However, it is clear that the tubulin band varies significantly across different samples, with an overall tendency to be higher in the macular regions. If tubulin and PHGDH vary independently across retinal regions, it cannot be a reasonably good loading control, thus the result in Figure 3D becomes somewhat questionable. Unfortunately, this is the critical data connecting cultured results with retinal tissues.

As noted in the comments for question 3 of the reviewer 1 above, for the western blot of total retinal protein, we loaded the same amount of total retinal protein (20ug) for each sample and normalized the data to total loaded protein amount; a similar finding was observed as shown in the revised Figure 5—figure supplement 1.

On mitochondria respiration measurements, judging from baseline data points before oligomycin in Figure 5G, it is difficult to relate it to the "basal respiration" data in Figure 5H, which shows a significant difference. Overall, the ECAR data looks much more convincing than the OCR data.

The basal respiration was derived by subtracting non-mitochondrial respiration from baseLine cellular OCR. Although the baseLine data points of M-hPMCs and P-hPMCs are similar before oligomycin, their non-mitochondrial respiration was significantly different after treatment with antimycin A and rotenone treatment, inhibitors of complex III and I used to shut down ETC function.